# Postprandial Apolipoprotein B48 is Associated with Subclinical Atherosclerosis in Patients with Rheumatoid Arthritis

**DOI:** 10.3390/jcm9082483

**Published:** 2020-08-02

**Authors:** Natalia Mena-Vázquez, Marta Rojas-Gimenez, Francisco Gabriel Jimenez Nuñez, Sara Manrique-Arija, José Rioja, Patricia Ruiz-Limón, Inmaculada Ureña, Manuel Castro-Cabezas, Pedro Valdivielso, Antonio Fernández-Nebro

**Affiliations:** 1The Institute of Biomedical Research in Malaga (IBIMA), 29010 Málaga, Spain; nataliamenavazquez@gmail.com (N.M.-V.); cortesfra@gmail.com (F.G.J.N.); sarama_82@hotmail.com (S.M.-A.); jose.rioja@uma.es (J.R.); patrilimon@hotmail.com (P.R.-L.); inuregar@gmail.com (I.U.); valdivielso@uma.es (P.V.); afnebro@gmail.com (A.F.-N.); 2UGC de Reumatología, Hospital Regional Universitario de Málaga, 29009 Málaga, Spain; 3UGC de Reumatología, Instituto Maimónides de Investigación Biomédica de Córdoba (IMIBIC), Hospital Universitario Reina Sofia, 14004 Córdoba, Spain; 4Departamento de Medicina y Dermatología, Universidad de Málaga, 29010, Málaga, Spain; 5Unidad de Gestión Clínica de Endocrinología y Nutrición, Hospital Clínico Virgen de la Victoria, 29010 Málaga, Spain; 6Department of Internal Medicine, Franciscus Gasthuis & Vlietland, Kleiweg 500, 3045 PM Rotterdam, The Netherlands; m.castrocabezas@franciscus.nl; 7UGC de Medicina Interna, Hospital Universitario Virgen de la Victoria, Universidad de Málaga, 29010 Málaga, Spain

**Keywords:** rheumatoid arthritis, postprandial lipemia, apolipoprotein B48, subclinical atherosclerosis

## Abstract

Objective: To describe postprandial lipemia in patients with rheumatoid arthritis (RA) and to analyze its association with subclinical atherosclerosis measured as carotid intima-media thickness (cIMT). Methods: We performed an observational study of 40 patients with RA and 40 sex and age-matched controls. Patients with dyslipidemia were excluded. Pathologically increased cIMT was defined as a carotid thickness greater than the 90th percentile (>p90) for age and sex. Fasting and postprandial plasma lipids, cholesterol, triglycerides, apolipoprotein B48 (ApoB48), and total ApoB were evaluated. The other variables included were clinical and laboratory values, Framingham score, and the 28-joint Disease Activity Score (DAS28). Two multivariate models were constructed to identify factors associated with pathologic cIMT in patients with RA. Results: Fasting lipid values were similar in patients with RA and controls, although those of postprandial ApoB48 were higher (median (IQR), 14.4 (10.8–12.1) vs. 12.1 (2.3–9,8); *p* = 0.042). Pathologic cIMT was recorded in 10 patients with RA (25%) and nine controls (22.5%). In patients with RA, pathologic cIMT was associated with postprandial ApoB48 (OR (95% CI), 1.15 (1.0–1.3)) and total ApoB (OR [95% CI], 1.12 [1.1–1.2]). The second model revealed a mean increase of 0.256 mm for cIMT in patients with elevated anticitrullinated protein antibodies (ACPAs). Conclusion: Postprandial ApoB48 levels in patients with RA are higher than in controls. Postprandial ApoB48 and total ApoB levels and markers of severity, such as ACPAs, are associated with pathologic cIMT in patients with RA. Our findings could indicate that these atherogenic particles have a negative effect on the endothelium.

## 1. Introduction

Rheumatoid arthritis (RA) is a chronic inflammatory disease characterized by persistent synovitis, bone erosions, and functional disability. It is associated with premature death and multiple morbidities [1], mainly because the cardiovascular risk of affected patients is similar to that of patients with type 2 diabetes mellitus [2]. Accelerated atherosclerosis in patients with RA is due to both the presence of traditional cardiovascular risk factors and to nontraditional cardiovascular risk factors, including systemic inflammation and dyslipidemia [3,4,5,6,7].

Ultrasound measurement of intima-media thickness (IMT) and the presence of plaques in the carotid artery has been reported to be a useful test for detecting subclinical atherosclerosis [8,9,10]. It has also been considered valid for confirming accelerated atherogenesis in persons with RA. In a study of a cohort of patients with chronic RA and no traditional cardiovascular risk factors or previous cardiovascular events, carotid IMT (cIMT) revealed abnormally high values compared with healthy controls [11].

More than half of patients with RA have dyslipidemia [12]—which is one of the reported factors that contributes to increased cardiovascular risk [13]. Park et al. [14] reported that in RA patients, levels of apolipoprotein A (ApoA) and HDL cholesterol were significantly lower than in controls. Similarly, lipoprotein levels, the ApoB/ApoA ratio, total cholesterol/HDL ratio, and LDL/HDL ratio were greater in untreated RA patients. In contrast with total cholesterol, LDL cholesterol, and HDL cholesterol, which are associated with cardiovascular risk, the importance of triglycerides as a cause of premature coronary disease is less clear [15,16]. Interest in triglycerides as a cardiovascular risk factor is growing because recent studies show that increased postprandial lipemia is an independent predictor of the risk of atherosclerosis in the general population [15,17]. Intolerance to triglycerides arises from the inability to process triglyceride-loaded lipoproteins, which in turn increases the risk of atherosclerosis [18,19]. For example, an association has been reported between higher ApoB48 levels and coronary atherosclerosis in patients with type 2 diabetes mellitus [20,21].

Few data have been published to date on subclinical atherosclerosis measured based on cIMT and its association with postprandial lipemia in patients with RA [7]. The objective of the present study was to analyze the association between postprandial lipid values (i.e., ApoB48 levels after a mixed breakfast) and subclinical atherosclerosis measured as cIMT and plaque presence in patients with RA. The results were compared with those of controls.

## 2. Patients and Methods

### 2.1. Design

We performed a controlled observational cross-sectional study of a cohort of patients with established RA and controls. The study was carried out at The Institute of Biomedical Research in Malaga (IBIMA) in the Rheumatology Department of Hospital Regional Universitario de Málaga (HRUM), in the Lipid and Atherosclerosis Laboratory of Universidad de Málaga (UMA) and the Center for Medical and Health Research (CIMES), Spain. The study was approved by the local Clinical Research Ethics Committee (CEIC) on 16/11/2018 (Code 1787-N-18). All participants gave their written informed consent.

### 2.2. Patients

Patients with RA were recruited consecutively between January and October 2019 from an initial cohort established between 2007 and 2012. The inclusion criteria were a diagnosis of RA (following the 2010 criteria of the American College of Rheumatology/European League Against Rheumatism [22]), onset after age 16 years, and being in prospective follow-up in the cohort at the cut-off date. The exclusion criteria were inflammatory diseases other than RA (except secondary Sjögren syndrome), diagnosis of dyslipidemia, lipid-lowering treatment, active infection, and pregnancy.

### 2.3. Controls

The controls were volunteers aged more than 16 years with no inflammatory or autoimmune diseases or symptoms that would lead us to suspect these diseases. The exclusion criteria were the same as for the patients. The controls were volunteers aged more than 16 years with no inflammatory or autoimmune diseases or symptoms that would lead us to suspect these diseases (i.e., joint inflammation, inflammatory pain, morning stiffness, etc.). The exclusion criteria were the same as for the patients. The controls were selected at random from a health center in the catchment area of the hospital. Controls and patients were matched by age and sex.

### 2.4. Protocol

A fasting preprandial blood sample was taken from all patients at 8:30 AM. The study population then had a mixed breakfast in the hospital cafeteria (milk, cured ham, cheese, olive oil, and bread). The nutritional information for breakfast was as follows—775 kcal, 50 g of fat (53% saturated fatty acids, 41% monounsaturated fatty acids, 6% polyunsaturated fatty acids), and 40 g of carbohydrates. During the postprandial period, the participants could only take their habitual medication (if they had it to hand) and water on demand. They had to rest and refrain from smoking. Four hours later, a second blood sample was taken and processed at the local laboratory, except for the lipid analysis sample, which was frozen immediately and then analyzed at the CIMES. On the same day, all participants underwent a physical examination, and their clinical history was taken according to a pre-established protocol. An ultrasound scan of the carotid artery was also taken. 

### 2.5. Main Outcome Measure

The main outcomes measures were ApoB48 postprandial lipid values and cIMT assessed using ultrasound. Pathologic cIMT was defined as a carotid thickness greater than the 90th percentile (>p90) for age and sex according to reference data for the Spanish population [23]. Atheromatous plaque was defined by consensus [24] as (1) focal thickening of the arterial wall protruding toward the lumen and measuring >0.5 mm, (2) more than 50% of the neighboring cIMT, or (3) cIMT >1.5 mm. The ultrasound scans of the carotid were taken using the ART.LAB system (ESAOTE, Barcelona, Spain) by a trained ultrasound specialist with experience in the technique.

Patients were classified into two groups based on the presence or absence of carotid plaque and/or pathologic cIMT.

### 2.6. Laboratory Measures

Plasma lipids, cholesterol, and triglycerides [25] were determined using enzymatic techniques (SPINREACT, Barcelona, Spain) in a Mindray BS 380 autoanalyzer (MINDRAY, Shenzhen, China). Chylomicrons (cholesterol and triglycerides) and very low-density lipoprotein (VLDL) were measured using sequential ultracentrifugation with the same enzymatic kits. HDL cholesterol was determined using a homogeneous assay (SIEMENS, Erlangen, Germany). LDL cholesterol was calculated based on the Friedewald formula. ApoB48 and total ApoB levels were measured in plasma after 8 h fasting and 4 h after the meal using a commercially available ELISA approach (Shibayagi Co Ltd., Ishihara, Japan) [20] and immunoturbidimetry (BIOSYSTEMS, Barcelona, Spain), respectively. Serum levels of high sensitivity C-reactive protein (hsCRP) were measured using turbidimetry (BIOSYSTEMS, Barcelona, Spain).

### 2.7. Other Variables

The other variables collected on the cut-off date included epidemiological variables (sex, race, body mass index (BMI), weight/height in m^2^) waist circumference (cm), hip circumference (cm), and the waist-hip ratio (quotient of waist circumference/hip circumference in cm)). We also collected conventional cardiovascular risk–related variables, as follows: Smoking (active, exsmoker, never), obesity (BMI > 30), arterial hypertension ≥140/90 mmHg or current antihypertensive medication [26], diabetes diagnosed according to the criteria of the American Diabetes Association [27], a personal history of cardiovascular disease, family history of coronary disease defined as a first-degree relative with myocardial infarction or stroke (<55 years for men and <60 years for women), Framingham score indicating risk of coronary heart disease at 10 years expressed as a percentage [28] (>20%, high-risk; 10–20%, intermediate risk; and <10%, low risk), and a validated questionnaire on adherence to a Mediterranean diet (MEDAS, 14 items [29], with ≥9 items considered adherent and <9 items considered nonadherent). The activity was assessed using the International Physical Activity Questionnaire (IPAQ) score [30], expressed as metabolic equivalent of task (MET) minutes, as follows: Low/sedentary or insufficient level of physical activity for the healthy activity recommendations, <600 MET minutes in the previous week; or moderate/high/fulfills the criteria for moderate levels, >600 MET minutes in the previous week.

The clinical-laboratory and therapeutic values recorded included rheumatoid factor (RF), which was considered positive if >10 IU/mL, and anticitrullinated protein antibody (ACPA), which was considered positive if >20 IU/mL. Inflammatory activity was evaluated in patients using the 28-joint Disease Activity Score with erythrocyte sedimentation rate (DAS28-ESR) (continuous, range, 0–9.4) [31] recorded at the visit [32]. According to the DAS28-ESR, activity was considered high with a value >5.1; moderate with 3.2–5.1; low with 2.6–3.2; and remission with ≤2.6. We also took into account severity variables, such as the presence of radiologic erosions and the score on the Health Assessment Questionnaire (HAQ) [32]. Treatment with conventional synthetic disease-modifying antirheumatic drugs (DMARDs) and biologic DMARDs was recorded.

### 2.8. Statistical Analysis

We used summary statistics to describe the participants and the χ^2^ and *t* or Mann-Whitney or Wilcoxon test to refute differences between them, as applicable.

We used summary statistics to describe the different variables. Normality was confirmed using the Kolmogorov-Smirnov test. The χ^2^ and t-test or Mann-Whitney test were used to compare the main characteristics between patients and controls, as well as between patients with RA and a normal and pathologic cIMT. Mean ranges for the baseline and postprandial values of ApoB48 and other lipid variables were compared between associated variables using the Wilcoxon test. Finally, we constructed two multivariate models: A backward binary logistic regression analysis (Wald) (dependent variable (DV): pathologic cIMT) and linear regression (DV: cIMT) to explore the variables that were independently associated with cIMT in patients with RA. The variables selected for the multivariate analysis that were significant in the bivariate analysis and those that were of clinical interest. Statistical significance was set at *p* < 0.05. The analyses were performed using R 2.4–0.

## 3. Results

### 3.1. Baseline Characteristics

The study population was comprised of 80 participants (40 patients with RA and 40 controls). Table 1 shows the baseline characteristics of both patients and controls. Most participants were women (85%), with a median age of around 56 years. Both groups were well balanced in terms of epidemiological characteristics, comorbidities, and cardiovascular risk factors. However, physical activity in MET minutes was better in the controls (median (interquartile range [IQR]) = 893 (280.0–1188.0) vs. 495.0 (70.0–990.0); *p* = 0.008)), who also had higher acute phase reactant and antibody levels.

Most patients were women with established RA (median (IQR) time since onset, 119 (81.2–167.9) months); 16/40 (40%) had erosive disease, and more than half were in remission or with low disease activity. A total of 31/40 (77.5%) were taking a synthetic DMARD, mainly methotrexate, and 21/40 (52.5%) were taking a biologic DMARD, mainly a tumor necrosis factor alpha (anti-TNF-α) drug. The median dose of corticosteroids at the date of the protocol was 5 mg of prednisone equivalents in 13/40 (32.5%) who were taking corticosteroids.

### 3.2. Study of Pre- and Postprandial Blood Lipid Values and cIMT in Patients and Controls

Table 2 shows the baseline and postprandial data for the lipid profile in patients and controls, as well as the findings for the carotid ultrasound.

In general, there were no differences between patients and controls with respect to fasting and postprandial blood lipid values in most readings. As for the postprandial increase in lipid values, a significant increase was recorded in triglycerides, chylomicrons (triglycerides), very-low-density lipoprotein (triglycerides), and ApoB48. The increase in ApoB48 was significantly greater in patients than in controls (median (IQR), 14.4 (10.8–23.2) vs. 12.1 (10.9–16.2); *p* = 0.042).

Ten patients with RA (25%) and nine controls (22.5%) had pathologic cIMT, although there were no significant differences in cIMT or in the number of plaques.

### 3.3. Study of pre- and Postprandial Blood Lipids and Baseline Characteristics of Patients with RA According to cIMT

As shown in Table 3, 10 of the 40 patients with RA (25%) had pathologic cIMT. There were fewer women in this group (*p* = 0.002), and the median waist-hip ratio was greater (*p* = 0.048), as was the median Framingham score (*p* = 0.039). There were no differences in the clinical and laboratory data, except that ACPAs were more frequently positive and with higher titters (>340) in patients with pathologic cIMT (*p* = 0.036). The correlation analysis did not show significant differences, although it shows a positive correlation with a tend statistical significance (r = 0.290; *p* = 0.096) (Figure 1). There were no differences in DMARDs, although patients with pathologic cIMT more frequently took glucocorticoids (*p* = 0.032).

With respect to preprandial blood lipids, patients with pathologic cIMT had higher levels of HDL cholesterol (*p* = 0.045), and triglycerides (*p* = 0.014) and higher levels of VLDL (triglycerides) (*p* = 0.022), and total cholesterol (*p* = 0.028) than patients with normal cIMT. Similar results were recorded for postprandial blood lipids, with lower levels of HDL cholesterol (*p* = 0.042) and higher levels of triglycerides (*p* = 0.033), chylomicrons (triglycerides) (*p* = 0.045), VLDL (triglycerides) (*p* = 0.036), and VLDL (cholsterol) (*p* = 0.046) in patients with pathologic cIMT (Figure 2). There were no preprandial differences in apolipoproteins, although postprandial ApoB48 was higher in patients with pathologic cIMT (*p* = 0.017). Furthermore, after a mixed breakfast, the increase in ApoB48 and chylomicrons (triglyceride) was significantly greater in patients with pathologic cIMT (*p* = 0.002 and *p* = 0.045, respectively) (Table 4). These differences were not found in the levels of blood lipids and ApoB48 between control group with pathologic cIMT and control group with normal cIMT (Appendix A).

### 3.4. Multivariate Analysis

Table 5 shows the results of the multivariate logistic regression analysis (DV: pathologic cIMT) in patients with RA. Total ApoB and postprandial ApoB48 were independently associated with pathologic cIMT in patients with RA, whereas female sex was a protective factor. Similarly, Table 6 shows an alternative multivariate linear regression analysis (DV: cIMT). In this model, cIMT continued to be independently associated with sex, total ApoB and postprandial ApoB48, and high ACPA titers (>340).

## 4. Discussion

Cardiovascular morbidity and mortality are high in patients with RA owing to accelerated atherosclerosis [33,34]. Hyperlipidemia is an increasingly interesting cardiovascular risk factor given that postprandial triglycerides are similar to and even better than fasting triglycerides as a predictor of cardiovascular disease, with the practical advantage that the patient is not required to fast [15,35,36]. Postprandial blood lipids have received little attention in patients with RA [37]. Therefore, we gave patients with RA and controls a mixed breakfast containing 50 g of different types of fat to determine the potential association with subclinical atherosclerosis measured using cIMT.

Consistent with this approach, we observed that fasting triglyceride and ApoB48 levels were similar in patients with RA and controls, although postprandial ApoB48 levels were higher. Burggraaf et al. [33] showed that patients with RA had higher levels of baseline ApoB48 than controls and that the accumulation of atherogenic chylomicron remnants could contribute to the high risk of cardiovascular disease in these patients. Similarly, the authors did not find differences in triglyceride levels between patients and controls, suggesting that lipolysis of chylomicrons could be normal in patients with RA and that perhaps the abnormality lies in the catabolism of chylomicron remnants in the liver.

Furthermore, we found that increased postprandial blood lipids in patients with RA was associated with pathologic cIMT. When both groups of patients were compared, we found that patients with pathologic cIMT had higher postprandial values for triglycerides, chylomicrons (triglycerides), VLDL (triglycerides), and ApoB48 than patients with normal cIMT. However, while baseline ApoB48 values were also higher in patients with pathologic cIMT, the difference was not statistically significant. This finding differs from those of a study of postprandial blood lipids in patients with type 2 diabetes mellitus [38], where higher postprandial and fasting ApoB48 levels were observed only in the subgroup with subclinical peripheral arterial disease. Our postprandial findings could be explained, as reported elsewhere, by the fact that postprandial triglycerides and ApoB48 are better associated with subclinical atherosclerosis in the carotid arteries [39] and femoral arteries [40]. Valero et al. [41] attempted to determine whether fasting ApoB48 level could replace postprandial blood lipids as a marker for evaluating the risk of coronary disease and found that fasting ApoB48 did not predict the risk of coronary disease, in contrast with postprandial levels.

An increasing number of studies show that patients with untreated active RA have lower levels of total cholesterol, LDL cholesterol, and HDL cholesterol [42]. This paradoxical association between lipids and RA has been associated with a greater risk of cardiovascular disease owing to current inflammation. Moreover, control of inflammation can increase these lipid values in serum [43]. In our study, we found no differences in cholesterol levels between patients and controls. However, while we found no differences in total cholesterol and LDL cholesterol between patients with pathologic or normal cIMT, the HDL value was lower in patients with pathologic cIMT. In this sense, some authors report an inverse relationship between HDL cholesterol and the presence of carotid plaque, but not cIMT [44], and attribute this relationship to an alteration in the cholesterol efflux capacity of HDL, thus potentially modifying inverse transport of cholesterol. Similarly, we did not find an association between HDL and pathologic cIMT, since HDL was eliminated from the multivariate model when we included sex, owing to the weight of this factor and to the higher number of men with pathologic cIMT (Appendix A). The differences found in apob48 between individuals with pathological and non-pathological cIMT were only found in the patient group and not in the control group, so this difference may be more related to the disease itself and not to other frequently associated factors with the lipids level.

After controlling for confounders in a multivariate logistic regression analysis, ApoB48, total ApoB, and sex were independent predictors of pathologic cIMT in patients with RA. As we have seen, some studies associate postprandial ApoB48 with pathologic cIMT. Postprandial chylomicrons and their remnants may have a direct effect on the development of atherosclerosis and an indirect effect by stimulating inflammation through activation of circulating leukocytes and the formation of foam cells [20,45,46,47]. We also found total ApoB to be a predictor of subclinical atherosclerosis in RA; this observation was to be expected, since all atherogenic lipoproteins carry ApoB100. Also, as expected, the multivariate analysis showed that men with RA had a greater risk of pathologic cIMT. This finding is consistent with the data reported by van Breukelen et al. [47], possibly because estrogens suppress progression of atherosclerosis and because other genetic and hormonal factors in men contribute to these differences [48].

The linear regression analysis also showed that postprandial ApoB48 and total ApoB levels, male sex, and high ACPA titers were associated with pathologic cIMT. This association with high levels of ACPA has been reported elsewhere [49,50,51]. Vázquez-Del Mercado et al. [51] demonstrated an independent association between ACPA values and cIMT, suggesting a possible role for ACPA in the pathogenesis of atherosclerosis in RA.

Our study is subject to a series of limitations. First, it is noteworthy that the cIMT values and the number of plaques were similar between patients with RA and controls, even though other studies have shown an increase in cIMT in patients with RA compared with controls, even at the onset of the disease [47]. This may be because our sample was not sufficiently large to show these differences and because of a bias resulting from the voluntary participation of the controls. It could also be due to the relatively low inflammatory activity of the disease in patients with RA, which may have had a beneficial effect on cIMT. Recent studies have shown a lower cardiovascular risk in RA cohorts, where a treat-to-target strategy, has been applied [52]. Furthermore, postprandial blood lipids are generally evaluated as an increase in the area under the curve for triglycerides after fat loading [53]; we measured ApoB48 as a measure of postprandial blood lipids. Nevertheless, we believe that our findings are reliable, because other studies have shown that measurement of ApoB48 4 h after a mixed breakfast correlates well with the area under the curve for triglycerides after 8 h [54]. In addition, the controls in our study were not completely healthy, since they had comorbidities that were common in the general population, such as arterial hypertension, although they did not have inflammatory disease or dyslipidemia and were not taking statins. However, this may make them similar to the study patients in terms of comorbidities not linked to RA itself. Guclocorticosteroids treatment was higher in patients with pathologic cIMT. However, this variable was eliminated from the multivariate model, probably due to the weight of lipid effect. In this sense, some authors have reported a relationship between glucocorticosteroids and cardiovascular events by increasing the risk, due to deleterious effects on lipids, among other causes [55,56]. Finally, although differences in postprandial ApoB48 between patients and controls were significative according to the design of our study, the weak significance of the results should be taken into account.

In conclusion, postprandial ApoB48 levels are higher in patients with RA than in controls. Similarly, our results support that postprandial levels of ApoB48, together with total ApoB levels, male sex, and factors affecting severity (e.g., ACPA), are associated with pathologic cIMT in patients with RA. While these findings should be compared in specific studies, they could point to delayed catabolism of chylomicrons, delayed uptake of chylomicrons in the liver, or more marked synthesis in patients with RA leading to a greater risk of atherosclerosis.

## Figures and Tables

**Figure 1 jcm-09-02483-f001:**
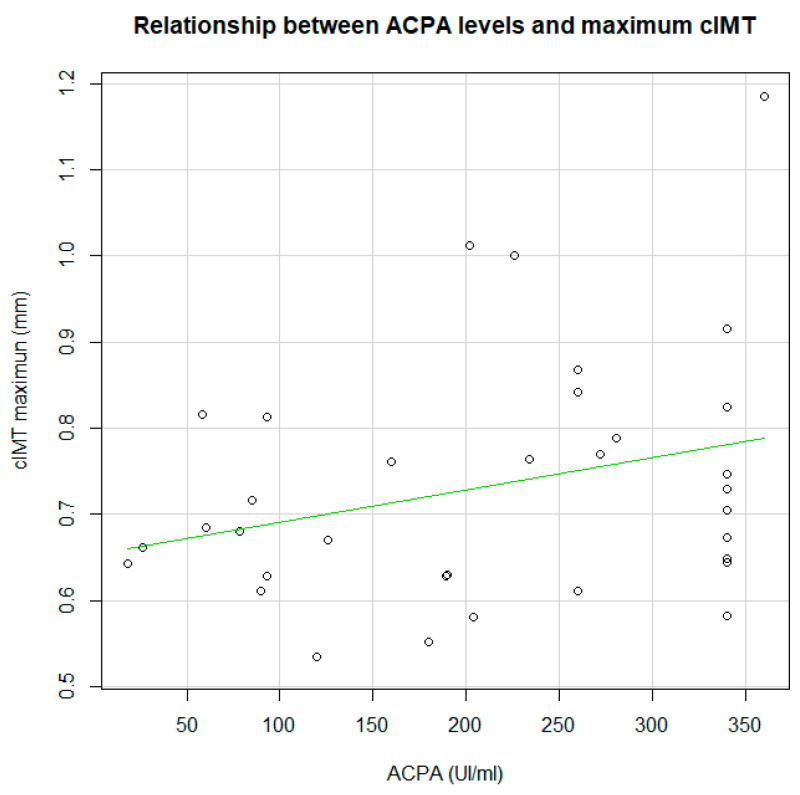
Correlation between anti-citrullinated peptide antibodies (ACPA)titters and carotid intima media thickness (cIMT) maximum.

**Figure 2 jcm-09-02483-f002:**
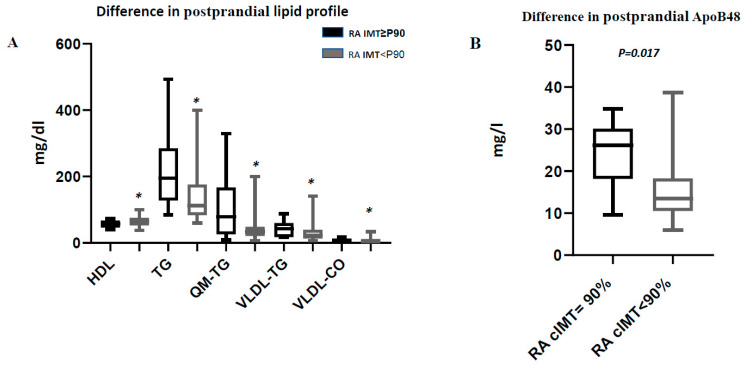
Postprandial lipid profile amongRA patients with cIMT≥90% and cIMT<90%. (**A**) Difference in postprandial lipid profile. * Indicates significant differences (*p*-value ≤ 0.05). (**B**) Difference in postprandial ApoB48. Abreviature; RA: Rheumatoid arthritis; cIMT:carotidintima-media thickness; HDL:high-densitylipoproteincholesterol; TG:Triglyceride; QM-TG: Triglyceridechylomicrons; VLDL-TG: VeryLow DensityLipoprotein-Triglyceride; VLDL-CO: VeryLow DensityLipoprotein-cholesterol.

**Table 1 jcm-09-02483-t001:** Baseline characteristics of 40 patients with RA and 40 controls.

Variable	Patients *n* = 40	Controls *n* = 40	*p* Value
**Epidemiological Characteristics**			
Age in Years, Median (IQR)	55.7 (52.9–61.7)	57.0 (52.7–61.1)	0.662
Female Sex; *n* (%)	35 (87.5)	34 (85.0)	0.745
Smoking			0.181
Never Smoked, *n* (%)	15 (37.5)	21 (52.5)	
Exsmoker, *n* (%)	19 (47.5)	11 (27.5)	
Active Smoker, *n* (%)	6 (15.0)	8 (20.0)	
Comorbidities			
Arterial Hypertension, *n* (%)	10 (25.0)	9 (22.5)	0.792
Diabetes Mellitus, *n* (%)	2 (5.0)	4 (10.0)	0.395
Cardiovascular Disease, *n* (%)	3 (7.5)	2 (5.0)	0.644
Family History of Coronary Artery Disease, *n* (%)	13 (32.5)	7 (17.5)	0.121
**Anthropometric Characteristics**			
BMI (kg/m^2^), Median (IQR)	26.7 (24.5–31.0)	27.2 (24.4–30.8)	0.758
Obesity, *n* (%)	10 (26.3)	10 (27.0)	0.944
Waist Circumference, (cm), Median (IQR)	91.5 (83.0–108.5)	91 (84.0–102.0)	0.361
Hip Circumference (cm), Median (IQR)	106.5 (103.5–112.3)	105.0 (100.0–114.0)	0.308
Waist-Hip Ratio, Median (IQR)	0.86 (0.8–0.9)	0.86 (0.8–0.9)	0.828
MET-Minute, Median (IQR)	495.0 (70.0–990.0)	893.0 (280.5–1188.0)	0.008
Total MEDAS Score, Median (IQR)	10.0 (8.0–11.0)	9.0 (8.0–11.0)	0.184
Framingham %, Median (IQR)	2.6 (0.9–4.1)	1.8 (0.7–4.6)	0.501
High Risk, *n* (%)	0 (0.0)	0 (0.0)	1.000
Intermediate Risk, *n* (%)	6 (16.2)	3 (8.1)	0.286
Low Risk, *n* (%)	31 (83.8)	34 (91.9)	0.286
**Clinical-Laboratory Characteristics**			
Progression of RA, Months, Median (IQR)	119 (81.2–167.9)	-	-
Diagnostic delay, months, median (IQR)	8.1 (5.6–16.7)	-	-
Erosions, *n* (%)	16 (40.0)	-	-
RF > 10, *n* (%)	26 (65.0)	0 (0.0)	<0.001
ACPA > 20, *n* (%)	31 (77.5)	0 (0.0)	<0.001
High-Sensitivity CRP (mg/dL), Median (IQR)	4.2 (2.7–7.4)	1.7 (0.8–3.1)	0.002
ESR (mm/h), Median (IQR)	15 (9.0–26.5)	11 (6.6–18.5)	0.016
DAS28 at Protocol, Median (IQR)	3.06 (2.5–4.2)	-	-
Remission-Low Activity, *n* (%)	21 (53.8)	-	-
Moderate-High Activity, *n* (%)	18 (46.1)	-	-
HAQ, Median (IQR)	0.9 (0.2–1.6)	-	-
Synthetic DMARDs, *n* (%)	31 (77.5)	-	-
Methotrexate, *n* (%)	23 (62.2)	-	-
Leflunomide, *n* (%)	3 (8.1)	-	-
Sulfasalazine, *n* (%)	3 (8.1)	-	-
Hydroxychloroquine, *n* (%)	2 (5.4)		
Biologic DMARDs, *n* (%)	21 (52.5)	-	-
Anti TNF-α, *n* (%)	17 (45.9)	-	-
Jak Inhibitor, *n* (%)	1 (2.7)	-	-
Anti-IL-6, *n* (%)	3 (8.1)	-	-
Glucocorticoid at Protocol, *n* (%)	13 (32.5)	-	-
Glucocorticoid Dose at Protocol, Median (IQR)	5 (5.0–5.0)	-	-
**Other Treatments**			
Antihypertensive Drugs	10 (25.0)	9 (22.5)	0.792
ACEIs, *n* (%)	7 (17.5)	7 (17.5)	0.778
ARAIIs, *n* (%)	3 (7.5)	2 (5.0)	0.462
Diuretics, *n* (%)	5 (12.5)	8 (20.0)	0.370
Metformin, *n* (%)	2 (5.0)	3 (7.5)	0.320
Insulin, *n* (%)	0 (0.0)	1 (2.5)	0.320
Other Oral Antidiabetic Agents, *n* (%)	0 (0.0)	1 (2.5)	0.320

Abbreviations: RA, rheumatoid arthritis; IQR, interquartile range; ACPA, anti-citrullinated peptide antibodies; RF, rheumatoid factor; SD, standard deviation; MEDAS, Mediterranean Diet Adherence Survey; DAS28, 28-joint Disease Activity Score; HAQ, Health Assessment Questionnaire; CRP, C-reactive protein; ESR, erythrocyte sedimentation rate; DMARD, disease-modifying antirheumatic drug; IL-6, interleukin 6; Anti TNF, anti–tumor necrosis factor ACEI, angiotensin-converting enzyme inhibitor; ARAII, angiotensin II receptor antagonists.

**Table 2 jcm-09-02483-t002:** Lipid profile and carotid ultrasound in 40 patients with RA and 40 controls.

Variable	RA *n* = 40		Controls *n* = 40		RA vs. Controls *p*	
	Fasting	Postprandial	Fasting	Postprandial	Fasting	Postprandial
**Fasting Lipid Profile**						
Total Cholesterol (mg/dL), Median (IQR)	212.1 (187.0–234.2)	202.0 (178.0–226.2)	200.2 (176.0–227.2)	201.0 (168.1–220.5)	0.148	0.222
LDL Cholesterol (mg/dL), Median (IQR)	127.0 (107.1–140.0)	110.3 (98.5–130.0)	116.5 (95.7–140.5)	108.0 (83.1–128.6)	0.229	0.203
HDL Cholesterol (mg/dL), Median (IQR)	62.5 (54.7–76.5)	62.2 (52.7–72.2)	59.5 (47.7–71)	57.1 (46.2–67.2)	0.162	0.063
Triglycerides (mg/dL), Median (IQR)	82.5 (66.7–113.5)	130.0 (91.7–185.0) *	88.5 (64.5–125.7)	132.5 (108.2–210.4) *	0.823	0.913
Chylomicrons (Triglycerides), Median (IQR)	14.7 (10.3–27.4)	42.3 (22.1–81.3) *	16.4 (7.5–39.8)	43.7 (31.9–84.7) *	0.644	0.225
Chylomicrons (Cholesterol), Median (IQR)	9.2 (6.8–13.5)	9.2 (6.8–13.5) *	12.3 (5.3–21.9)	12.3 (5.3–21.9) *	0.544	0.613
VLDL (Triglycerides), Median (IQR)	16.0 (10.1–27.7)	29.6 (15.5–41.1) *	21.0 (9.8–31.5)	24.6 (17.6–38.7) *	0.758	0.859
VLDL (Cholesterol), Median (IQR)	3.6 (2.3–6.1)	3.6 (2.3–6.1) *	5.8 (2.7–9.8)	5.8 (2.7–9.8) *	0.087	0.083
ApoB48, Median (IQR)	7.4 (6.2–10.5)	14.4 (10.8–23.2) *	7.7 (5.5–10.3)	12.1 (10.9–16.2) *	0.874	0.042
ApoB Total, Median (IQR)	96.1 (84.9–104.9)	92.4 (80–103.4)	98.0 (84.3–108.3)	93.2 (77.3–102.9)	0.950	0.517
TG/HDL Ratio, Median (IQR)	1.2 (0.8–2.0)	2.1 (1.3–3.5)	1.4 (0.8–2.9)	2.5 (1.6–4.1)	0.597	0.574
ApoB48/TG Ratio, Median (IQR)	0.09 (0.07–0.1)	0.1 (0.7–0.1)	0.08 (0.1–0.13)	0.09 (0.7–0.1)	0.985	0.326
**Fasting Carbohydrate Profile**						
Baseline Blood Sugar (mg/dL), Median (IQR)	78.0 (74.7–83)		80.0 (72.7–88.2)		0.843	0.277
Homocysteine, Median (IQR)	14.4 (12.8–18)		13.5 (11.4–16.8)		0.494	
**Carotid Ultrasound**						
Pathologic cIMT >p90, *n* (%)	10 (25.0)		9 (22.5)		0.555	
Right cIMT (mm), Median (IQR)	0.7 (0.6–0.8)		0.7 (0.7–1.0)		0.652	
Left cIMT (mm), Median (IQR)	0.66 (0.6–0.7)		0.7 (0.64–0.78)		0.353	
Patients with Atheromatous Plaques, *n* (%)	7 (18.4)		8 (20.0)		0.481	

* *p* < 0.005 fasting vs. postprandial value. Abbreviations IQR, interquartile range; cIMT, carotid intima media thickness; LDL, low-density lipoprotein; HDL, high-density lipoprotein; TG, triglycerides; VLDL, very-low-density lipoproteins.

**Table 3 jcm-09-02483-t003:** Baseline characteristics of patients with RA according to cIMT.

Variable	RA with IMT >p90 *n* = 10	RA with IMT ≤p90 *n* = 30	*p* Value
Age, Years, Median (IQR)	55.3 (48.6–68.3)	55.7 (53.3–61.6)	0.900
Female Sex; *n* (%)	6 (60.0)	29 (96.7)	0.002
Smoking			0.818
Never, *n* (%)	4 (40.0)	11 (36.7)	
Exsmoker, *n* (%)	4 (40.0)	15 (50.0)	
Active Smoker, *n* (%)	2 (20.0)	4 (13.3)	
**Comorbidities**			
Arterial Hypertension, *n* (%)	3 (30.0)	7 (23.3)	0.673
Diabetes Mellitus, *n* (%)	1 (10.0)	1 (3.3)	0.442
Cardiovascular Disease, *n* (%)	0 (0.0)	3 (10.0)	0.298
**Anthropometric Characteristics**			
BMI (kg/m^2^), Median (IQR)	27.1 (24.4–32.2)	26.6 (24.6–29.5)	0.700
Obesity, *n* (%)			
Waist Circumference, (cm), Median (IQR)	106.5 (87–110.7)	89 (83–103)	0.212
Hip Circumference (cm), Median (IQR)	106 (103.2–109.7)	106.5 (102.2–112.2)	0.941
Waist-Hip Index, Median (IQR)	0.92 (0.83–1)	0.85 (0.81–0.91)	0.048
MET-Minute, Median (IQR)	247.5 (70.0–618.7)	594.0 (84.0–1064.0)	0.164
Total MEDAS, Median (IQR)	10. (8.7–11.0)	10.0 (8.0–11.0)	0.824
Framingham %, Median (IQR)	4.6 (1.5–13.8)	1.2 (0.6–3.8)	0.039
**Clinical-Laboratory Characteristics**			
Time Since Diagnosis of RA, Months, Median (IQR)	140 (93–214.4)	113 (80–166.2)	0.138
Diagnostic Delay, Months, Median (IQR)	9.9 (5.5–18.5)	6.9 (5.3–12.0)	0.414
Erosions, *n* (%)	4 (40.0)	12 (40.0)	0.473
RF > 10, *n* (%)	7 (70.0)	21 (70.0)	1.000
ACPA > 20, *n* (%)	8 (80.0)	22 (73.3)	0.473
High ACPA (>340), *n* (%)	6 (60.0)	10 (33.3)	0.036
High-Sensitivity CRP (mg/dL), Median (IQR)	4.4 (3.2–8.7)	3.8 (2.5–7.5)	0.221
ESR (mm/h), Median (IQR)	12.0 (7.7–37.2)	15.0 (9.0–26.0)	0.839
DAS28 at Protocol, Median (IQR)	3.3 (2.5–3.9)	2.9 (2.5–4.2)	0.644
HAQ, Median (IQR)	1.3 (0.7–1.7)	0.8 (0.2–1.6)	0.544
Synthetic DMARDs, *n* (%)	9 (90.0)	22 (73.0)	0.174
Methotrexate, *n* (%)	5 (50.0)	18 (60.0)	0.580
Biologic DMARDs, *n* (%)	4 (40.0)	18 (60.0)	0.271
Corticosteroids, *n* (%)	6 (60.0)	7 (23.3)	0.032

Abbreviations: cIMT, carotid intima media thickness; IQR, interquartile range; BMI, body mass index.

**Table 4 jcm-09-02483-t004:** Lipid profile and carotid ultrasound in RA patients according to cIMT.

Variable	RA with IMT >p90 *n* = 10		RA with IMT ≤p90 *n* = 30		RA with IMT >p90 vs. RA with IMT ≤p90 *p*	
	Fasting	Postprandial	Fasting	Postprandial	Fasting	Postprandial
**Fasting Lipid Profile**						
Total Cholesterol (mg/dL), Median (IQR)	226.0 (189.5–243.2)	204.1 (184–237.2)	210.2 (187.2–225.7)	200.0 (179.2–222.2)	0.471	0.573
LDL Cholesterol (mg/dL), Median (IQR)	138.2 (118.7–152.0)	110.0 (107.2–130.5)	123.5 (105.5–139.0)	110.0 (95.5–129.7)	0.266	0.647
HDL Cholesterol (mg/dL), Median (IQR)	61.0 (51.2–69.7)	57.5 (49.0–65.7)	66.5 (56.5–77.5)	63.5 (53.7–73.7)	0.045	0.042
Triglycerides (mg/dL), Median (IQR)	112.0 (82.7–17.6)	195.0 (127.5–285.2)	77.5 (863.5–107.2)	116.0 (83.5–176.2)	0.014	0.033
Chylomicrons (Triglycerides), Median (IQR)	33.8 (14.2–57.0) *	79.1 (25.6–167.1) *	14.1 (9.3–23.2)	32.7 (21.8–54.7)	0.066	0.045
Chylomicrons (Cholesterol), Median (IQR)	12.3 (8.6–16.1)	15.6 (5.7–25.0)	8.7 (6.0–11.6)	15.3 (7.3–24.5)	0.089	0.770
VLDL (Triglycerides), Median (IQR)	24.2 (21.9–39.5)	42.6 (17.3–60.0)	14.4 (9.8–26.4)	23.1 (13.1–39.7)	0.022	0.036
VLDL (Cholesterol), Median (IQR)	5.3 (3.9–9.4)	8.2 (3.7–12.7)	2.8 (1.9–6.0)	3.9 (2.7–8.9)	0.028	0.046
ApoB48, Median (IQR)	8.5 (5.9–13.0) *	24.3 (15.1–27.1) *	7.7 (5.5–10.3)	13.5 (10.5–18.2)	0.186	0.017
ApoB Total, Median (IQR)	102.7 (94.4–115.1)	98.0 (87–110.8)	94 (82.6–104.3)	91.7 (76.1–102.6)	0.141	0.100
**Increased Postprandial Blood Lipids**						
Triglycerides (mg/dL), Median (IQR)		73.2 (24.0–134.5)		39.9 (17.2–68.0)		0.122
Chylomicrons (Triglycerides), Median (IQR)		47.4 (14.6–124.1)		21.5 (10.2–37.7)		0.045
VLDL (Triglycerides), Median (IQR)		12.9 (6.4–20.7)		8.0 (3.0–16.2)		0.424
ApoB48, median (IQR)		12.3 (10.8–14.3)		6.7 (3.4–8.6)		0.002

* *p* < 0.005 fasting vs. postprandial value. Abbreviations: cIMT, carotid intima media thickness; LDL, low-density lipoprotein; HDL, high-density lipoprotein; TG, triglycerides; VLDL, very-low-density lipoprotein.

**Table 5 jcm-09-02483-t005:** Logistic regression of characteristics associated with pathologic cIMT (p > 90) in patients with rheumatoid arthritis.

Predictor	OR	95% CI	*p* Value
Female Sex	0.010	0.000–0.381	0.014
Postprandial ApoB48 *	1.159	1.021–1.315	0.023
Total ApoB	1.121	1.109–1.259	0.046

Nagelkerke R^2^ = 0.450, Variables not included in the equation: age, HDL post, TG post, ACPA >340, Framingham, smoking, corticosteroids * ApoB48 log transformed. Abbreviations; CI, confidence interval.

**Table 6 jcm-09-02483-t006:** Multiple linear regression of characteristics associated with cIMT in patients with RA.

Dependent Variable	Predictor	B	95% CI for B	*p* Value
Pathologic cIMT	Female Sex	−0.607	−0.306 to −0.151	<0.001
	Postprandial ApoB48	0.285	0.002 to 0.013	0.002
	Total ApoB	0.239	0.001 to 0.005	0.047
	ACPA≥340	0.256	0.018 to 0.137	0.018

Nagelkerke R^2^ = 0,520, Variables not included in the equation: age, HDL post, TG post, Framingham, smoking, corticosteroids. Abbreviations; CI, confidence interval; B, beta coefficient.

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
