# Peer review of "Postprandial Apolipoprotein B48 is Associated with Subclinical Atherosclerosis in Patients with Rheumatoid Arthritis"

_jcm, 2020, doi:10.3390/jcm9082483_

Round 1

Reviewer 1 Report

The concept studied, postprandial lipidemia and apoB-impact is interesting and original, not so much has been done in this field in rheumatology and RA. 

The age of the patients and controls has to be clarified, is it 59 (patients) or 56 as is stated somewhere. If there is about 10% age difference between case and controls which is written in table 1, this could have big impact on conclusions since postprandial effects usually are associated with age. I guess there is an error in the. table, but this must be clarified. 

The association reported is weak, borderline significant actually, and the discussion should be a little more clear about this. 

Author Response

Reviewer #1: 

The concept studied, postprandial lipidemia and apoB-impact is interesting and original, not so much has been done in this field in rheumatology and RA. 

The age of the patients and controls has to be clarified, is it 59 (patients) or 56 as is stated somewhere. If there is about 10% age difference between case and controls which is written in table 1, this could have big impact on conclusions since postprandial effects usually are associated with age. I guess there is an error in the. table, but this must be clarified. 

R:  Thanks for the comment. It is certainly a digitization error in the table. The correct age of the cases is 55.8 (11.4) as it is correctly written into the text. We revised the rest of the table in order to find any other error but everything else is ok. Finally, this variable has been expressed as median at the request of another reviewer:

Table 1. Baseline characteristics of 40 patients with RA and 40 controls.

Variable

Patients n=40

Controls n=40

p Value

Epidemiological characteristics

Age in years, median (IQR)

55.8 (52.9-61.7)

57.0 (52.7-61.1)

0.662

The association reported is weak, borderline significant actually, and the discussion should be a little clearer about this. 

R: We agree with your comment, and it is probably due to our sample size.  We added this limitation to the discussion.

Line 387-389: “Finally, although differences in postprandial ApoB48 between patients and controls were significative according to the design of our study, the weak significance of the results should be taken into account.”

Reviewer 2 Report

The work of Mena-Vázquez N. et al analyzed cardiovascular risk factors in RA patients and they found that the apolipoprotein B48 is associated with the cIMT. The work is properly designed and the results clearly presented. However I have some concerns related with the interpretation of the results and I miss some analysis.

Major points

  1. The authors found differences in the levels of blood lipids and ApoB48 between RA patients with pathologic cIMT and patients with normal pathologic cIMT. But are these differences also found in the control group? This is an important point because it will determine if these differences are specific of the disease.

  2. Related to this, do the authors find these differences if they exclude the men from the comparations? Because as they explain in the discussion the gender has a great influence in the cardiovascular risk and therefore the higher risk may be only due to a higher frequency of men in the pathologic cIMT group.

  3. The authors should also discuss in more detail the effect of glucocorticoids, as the use of them has been associated with a higher cardiovascular risk in RA patients (Roubille C, Richer V, Starnino T, et al The effects of tumour necrosis factor inhibitors, methotrexate, non-steroidal anti-inflammatory drugs and corticosteroids on cardiovascular events in rheumatoid arthritis, psoriasis and psoriatic arthritis: a systematic review and meta-analysis. Annals of the Rheumatic Diseases 2015;74:480-489).

Minor points

  1. ACPAs levels were higher in patients with pathologic cIMT (p=0.036). I recommend showing the correlation analysis.

  2. In table 1 the VLDL are exactly the same in both control and RA groups in the fasting and postprandial. Is that correct or there is an error in the table?

  3. In all the tables the authors should include the IQR instead the SD.

  4. Why secondary Sjögren syndrome was not an exclusion criteria?

Author Response

Reviewer #2: 

Major points

  1. The authors found differences in the levels of blood lipids and ApoB48 between RA patients with pathologic cIMT and patients with normal pathologic cIMT. But are these differences also found in the control group? This is an important point because it will determine if these differences are specific of the disease.

R:  These differences were not found in the levels of blood lipids and ApoB48 between control group with pathologic cIMT and control group with normal cIMT.  The results in the control group are shown in the following table (we added this table in supplementary material).

We have added in results, line 254-256: These differences were not found in the levels of blood lipids and ApoB48 between control group with pathologic cIMT and control group with normal cIMT (supplementary material).

We have added in the discussion, line 341-344: “The differences found in apob48 between individuals with pathological and non-pathological cIMT were only found in the patient group and not in the control group, so this difference may be more related to the disease itself and not to other frequently associated factors with the lipids level”.

Supplementary Table 1. Lipid profile and carotid ultrasound in control group according to cIMT.

Variable

Control group with IMT >p90 n=9

Control group with IMT ≤p90

n=31

 Control group with IMT >p90 vs with IMT ≤p90

p-value

Fasting

Postprandial

Fasting

Postprandial

Fasting

Postprandial

Fasting lipid profile

   Total cholesterol (mg/dl), median (IQR)

190.0 (171.1-226.5)

189.0 (166.5-220.5)

202.0 (176.0-236.0)

201.5 (167.7-227.0)

0.656

0.678

   LDL cholesterol (mg/dl), median (IQR)

110.0 (83.5-148.0)

108.0 (75.5-131.5)

120.0 (96.0-142.0)

108.5 (83.0-128.0)

0.667

0.638

   HDL cholesterol (mg/dl), median (IQR)

56.0 (45.5-74.0)

53.0 (46.5-68.5)

60.0 (48.0-71.0)

58.0 (45.0-66.0)

0.762

0.830

   Triglycerides (mg/dl), median (IQR)

72.0 (54.5-140.5)

116.5 (67.7-206.7)

86.0 (60.0-128.0)

136.0 (97.2-170.0)

0.545

0.473

   Chylomicrons (triglycerides), median (IQR)

15.2 (5.3-42.9)

 51.2 (15.3-78.2)

17.0 (7.5-45.3)

 54.3 (32.4-80.7)

0.831

0.473

   Chylomicrons (cholesterol), median (IQR)

5.5 (2.6-22.4)

29.1 (5.5-36.8)

11.0 (7.5-45.3)

14.1 (8.7-32.9)

0.509

0.497

   VLDL (triglycerides), median (IQR)

13.6 (8.5-43.7)

24.2 (11.2-57.1)

20.7 (9.8-31.1)

25.0 (17.9-38.1)

0.935

0.911

   VLDL (cholesterol), median (IQR)

5.9 (1.6-14.1)

7.1 (4.0-15.6)

5.7 (2.8-8.3)

7.4 (3.9-10.6)

0.730

0.543

   ApoB48, median (IQR)

5.2 (4.6-8.0)

14.4 (9.4-15.6)

8.3 (6.4-11.8)

13.9 (11.7-17.2)

0.116

0.274

   ApoB total, median (IQR)

101.9 (80.0-108.0)

94.1 (74.0-105.1)

96.8 (82.4-111.2

92.9 (76.7-104.0)

0.975

0.846

Increased postprandial blood lipids

   Triglycerides (mg/dl), median (IQR)

42.7 (16.1-128.8)

48.4 (32.2-75.8)

0.709

   Chylomicrons (triglycerides), median (IQR)

29.3 (11.8-135.3)

33.5 (13.1-64.0)

0.289

   VLDL (triglycerides), median (IQR)

9.8 (2.2-22.4)

8.4 (2.0-17.7)

0.975

   ApoB48, median (IQR)

6.7 (3.3-9.9)

6.9 (2.2-9.8)

0.920

* p <0.005 fasting vs postprandial value. Abbreviations: cIMT, carotid intima media thickness; LDL, low-density lipoprotein; HDL, high-density lipoprotein; TG, triglycerides; VLDL, very-low-density lipoprotein.

  1. Related to this, do the authors find these differences if they exclude the men from the comparations? Because as they explain in the discussion the gender has a great influence in the cardiovascular risk and therefore the higher risk may be only due to a higher frequency of men in the pathologic cIMT group.

R:  The results obtained are similar when we excluded men from the comparations, except for the HDL values. The patients with pathologic cIMT did not have higher levels of HDL cholesterol. However, an increase in lipids and ApoB48 was observed between RA patients with pathologic cIMT and patients with normal pathologic cIMT. 

With respect to pre-prandial blood lipids, patients with pathologic cIMT had higher levels of triglycerides (p=0.021) and higher levels of VLDL (triglycerides) (p=0.011), and Chylomicrons (triglycerides)(p=0.011) than the patients with normal cIMT. Similar results were recorded for postprandial blood lipids, with higher levels of triglycerides (p=0.042), chylomicrons (triglycerides) (p=0.052), and VLDL (cholesterol) (p=0.048) in patients with pathologic cIMT. There were no preprandial differences in apolipoproteins, although postprandial ApoB48 was higher in patients with pathologic cIMT (p=0.046). Furthermore, after a mixed breakfast, the increase in ApoB48 and chylomicrons (triglyceride) was significantly greater in patients with pathologic cIMT (p=0.032 and p=0.042, respectively). These results are shown in the following table for the reviewer.

We added in the text line 338-341: Similarly, we did not find an association between HDL and pathologic cIMT, since HDL was eliminated from the multivariate model when we included sex, owing to the weight of this factor and to the higher number of men with pathologic cIMT (supplementary material).

We added the next table to supplementary material:

Supplementary Table 2. Lipid profile and carotid ultrasound in RA patients according to cIMT excluded the men.

Variable

RA with IMT >p90 n=6

RA with IMT ≤p90

n=29

 RA with IMT >p90 vs RA with IMT ≤p90

p

Fasting

Postprandial

Fasting

Postprandial

Fasting

Postprandial

Fasting lipid profile

   Total cholesterol (mg/dl), median (IQR)

234.5 (212.7-255.7)

210.0 (195.2-252.5)

212.0 (185.5-227.5)

200.0 (177.0-228.0)

0.176

0.235

   LDL cholesterol (mg/dl), median (IQR)

139.0 (133.0-165.5)

116.0 (105.7-145.0)

122.0 (103.5-140.5)

110.0 (94.0-129.5)

0.164

0.312

   HDL cholesterol (mg/dl), median (IQR)

66.5 (60.2-78.2)

62.5 (55.7-72.2)

66.0 (53.5-80.5)

62.0 (52.5-77.0)

0.749

0.783

   Triglycerides (mg/dl), median (IQR)

112.0 (83.5-165.7)

195.0 (125.2-326.5)

74.0 (63.0-105.0)

111.0 (83.0-163.5)

0.021

0.042

   Chylomicrons (triglycerides), median (IQR)

52.7 (16.6-64.9) *

77.7 (22.9-195.4) *

13.8 (8.7-20.9)

34.2 (21.7-48.9)

0.011

0.052

   Chylomicrons (cholesterol), median (IQR)

13.4 (8.0-23.3)

15.5 (5.7-33.2)

8.5 (6.0-10.7)

14.3 (7.2-23.2)

0.093

0.782

   VLDL (triglycerides), median (IQR)

27.2 (21.9-42.4)

37.3 (17.3-60.0)

13.8 (9.7-23.1)

20.7 (12.9-38.7)

0.011

0.098

   VLDL (cholesterol), median (IQR)

5.9 (4.9-10.7)

8.8 (3.4-12.7)

2.7 (1.9-5.4)

3.9 (2.6-8.9)

0.014

0.048

   ApoB48, median (IQR)

7.5 (5.5-12.1) *

23.7 (10.0-31.1) *

7.0 (6.1-10.2)

13.4 (10.4-18.4)

0.685

0.046

   ApoB total, median (IQR)

104.0 (98.9-120.8)

106.3 (89.8-114.0)

94.5 (79.2-105.5)

91.9 (75.3-103.0)

0.093

0.062

Increased postprandial blood lipids

   Triglycerides (mg/dl), median (IQR)

63.7 (16.2-140.0)

35.7 (17.1-61.6)

0.279

   Chylomicrons (triglycerides), median (IQR)

46.4 (11.0-147.0)

20.0 (10.2-34.7)

0.042

   VLDL (triglycerides), median (IQR)

11.8 (6.4-22.0)

7.0 (3.0-16.3)

0.454

   ApoB48, median (IQR)

13.0 (6.2-16.4)

6.7 (3.4-8.4)

0.032

* p <0.005 fasting vs postprandial value. Abbreviations: cIMT, carotid intima media thickness; LDL, low-density lipoprotein; HDL, high-density lipoprotein; TG, triglycerides; VLDL, very-low-density lipoprotein.

  1. The authors should also discuss in more detail the effect of glucocorticoids, as the use of them has been associated with a higher cardiovascular risk in RA patients (Roubille C, Richer V, Starnino T, et al The effects of tumour necrosis factor inhibitors, methotrexate, non-steroidal anti-inflammatory drugs and corticosteroids on cardiovascular events in rheumatoid arthritis, psoriasis and psoriatic arthritis: a systematic review and meta-analysis. Annals of the Rheumatic Diseases 2015;74:480-489).

R: We have included the glucocorticosteroids variable in multivariate logistic regression analysis (dependent variable [DV]: pathologic cIMT) and similarly in alternative multivariate linear regression analysis (DV: cIMT). We have changed table 5 and table 6. However, this study was not designed to assess the effect of drugs, including glucocorticoids, on cIMT

Table 5. Logistic regression of characteristics associated with pathologic cIMT (p>90) in patients with rheumatoid arthritis.

Predictor

OR

95% CI

p Value

Female sex

0.010

0.000 – 0.381

0.014

Postprandial ApoB48*

1.159

1.021- 1.315

0.023

Total ApoB

           1.121

          1.109-1.259

        0.046

Nagelkerke R2 = 0.450, Variables not included in the equation: age, HDL post, TG post, ACPA >340, Framingham, smoking, glucocorticosteroids. *ApoB48 log transformed.

Table 6. Multiple linear regression of characteristics associated with cIMT in patients with RA.

Dependent variable

Predictor

B

95% CI for B

p Value

Pathologic cIMT

Female sex

–0.607

–0.306 to –0.151

<0.001

Postprandial ApoB48

0.285

0.002 to 0.013

0.002

Total ApoB

0.239

0.001 to 0.005

0.047

ACPA≥340

0.256

0.018 to 0.137

0.018

Nagelkerke R2 = 0,520, Variables not included in the equation: age, HDL post, TG post, Framingham, smoking, and glucocorticosteroids.

We added in the text, line 383-387:  Corticosteroids treatment was higher in patients with pathologic cIMT. However, this variable was eliminated from the multivariate model, probably due to the weight of lipid effect. In this sense, some authors have reported a relationship between glucocorticosteroids and cardiovascular events by increasing the risk due to deleterious effects on lipids, among other causes (1, 2).

We added references, line 554-560:

  1. Roubille C, Richer V, Starnino T, McCourt C, McFarlane A, Fleming P, et al. The effects of tumour necrosis factor inhibitors, methotrexate, non-steroidal anti-inflammatory drugs and corticosteroids on cardiovascular events in rheumatoid arthritis, psoriasis and psoriatic arthritis: a systematic review and meta-analysis. Ann Rheum Dis. 2015;74(3):480-9.
  2. Panoulas VF, Douglas KM, Stavropoulos-Kalinoglou A, Metsios GS, Nightingale P, Kita MD, et al. Long-term exposure to medium-dose glucocorticoid therapy associates with hypertension in patients with rheumatoid arthritis. Rheumatology (Oxford). 2008;47(1):72-5.

Minor points

  1. ACPAs levels were higher in patients with pathologic cIMT (p=0.036). I recommend showing the correlation analysis.

R: When analyzing the ACPA levels with the maximum GIMc as a quantitative variable, the correlation analysis did not show significant differences although it shows a positive correlation with a tend statistical significance (r= 0.290; p= 0.096).

We added in the text, line 223-227: There were no differences in the clinical and laboratory data, except that ACPAs were more frequently positive and with higher titters (>340) in patients with pathologic cIMT (p=0.036). The correlation analysis did not show significant differences although it shows a positive correlation with a tend statistical significance (r= 0.290; p= 0.096) (Figure 1). There were no differences in DMARDs, although patients with pathologic cIMT more frequently took glucocorticoids (p=0.032).

We added Figure 1:

  1. In table 1 the VLDL are exactly the same in both control and RA groups in the fasting and postprandial. Is that correct or there is an error in the table?

R: Thanks for the comment. It is certainly a digitization error in the table.  We have changed in table 2 median and p-value.

Table 2. Lipid profile and carotid ultrasound in 40 patients with RA and 40 controls.

Variable

RA

n=40

Controls

n=40

RA vs Controls

p

Fasting

Postprandial

Fasting

Postprandial

Fasting

Postprandial

Fasting lipid profile

   Total cholesterol (mg/dl), median (IQR)

212.1 (187.0-234.2)

202.0 (178.0-226.2)

200.2 (176.0-227.2)

201.0 (168.1-220.5)

0.148

0.222

   LDL cholesterol (mg/dl), median (IQR)

127.0 (107.1-140.0)

110.3 (98.5-130.0)

116.5 (95.7-140.5)

108.0 (83.1-128.6)

0.229

0.203

   HDL cholesterol (mg/dl), median (IQR)

62.5 (54.7-76.5)

62.2 (52.7-72.2)

59.5 (47.7-71)

57.1 (46.2-67.2)

0.162

0.063

   Triglycerides (mg/dl), median (IQR)

82.5 (66.7-113.5)

130.0 (91.7-185.0) *

88.5 (64.5-125.7)

132.5 (108.2-210.4) *

0.823

0.913

   Chylomicrons (triglycerides), median (IQR)

14.7 (10.3-27.4)

42.3 (22.1-81.3) *

16.4 (7.5-39.8)

43.7 (31.9-84.7) *

0.644

0.225

   Chylomicrons (cholesterol), median (IQR)

9.2 (6.8-13.5)

9.2 (6.8-13.5) *

12.3 (5.3-21.9)

12.3 (5.3-21.9) *

0.544

0.613

   VLDL (triglycerides), median (IQR)

16.0 (10.1-27.7)

29.6 (15.5-41.1) *

21.0 (9.8-31.5)

24.6 (17.6-38.7) *

0.758

0.859

   VLDL (cholesterol), median (IQR)

3.6 (2.3-6.1)

3.6 (2.3-6.1) *

5.8 (2.7-9.8)

5.8 (2.7-9.8) *

0.087

0.083

   ApoB48, median (IQR)

7.4 (6.2-10.5)

14.4 (10.8-23.2) *

7.7 (5.5- 10.3)

12.1 (10.9-16.2) *

0.874

0.042

   ApoB total, median (IQR)

96.1 (84.9-104.9)

92.4 (80-103.4)

98.0 (84.3-108.3)

93.2 (77.3-102.9)

0.950

0.517

   TG/HDL ratio, median (IQR)

1.2 (0.8-2.0)

2.1 (1.3-3.5)

1.4 (0.8-2.9)

2.5 (1.6-4.1)

0.597

0.574

   ApoB48/TG ratio, median (IQR)

0.09 (0.07-0.1)

0.1 (0.7-0.1)

0.08 (0.1-0.13)

0.09 (0.7-0.1)

0.985

0.326

Fasting carbohydrate profile

   Baseline blood sugar (mg/dl), median (IQR)

78.0 (74.7-83)

80.0 (72.7-88.2)

0.843

0.277

   Homocysteine, median (IQR)

14.4 (12.8-18)

13.5 (11.4- 16.8)

0.494

Carotid ultrasound

Pathologic cIMT >p90, n (%)

10 (25.0)

9 (22.5)

0.555

Right cIMT (mm), median (IQR)

0.7 (0.6-0.8)

0.7 (0.7-1.0)

0.652

Left cIMT (mm), median (IQR)

0.66 (0.6-0.7)

0.7 (0.64-0.78)

0.353

Patients with atheromatous plaques, n (%)

7 (18.4)

8 (20.0)

0.481

* p <0.005 fasting vs postprandial value. Abbreviations cIMT, carotid intima media thickness; LDL, low-density lipoprotein; HDL, high-density lipoprotein; TG, triglycerides; VLDL, very-low-density lipoproteins.

  1. In all the tables the authors should include the IQR instead the SD.

R: Following the reviewer's recommendations, we have changed all the tables with median (IQR) instead the mean (SD).

Table 1. Baseline characteristics of 40 patients with RA and 40 controls.

Variable

Patients n=40

Controls n=40

p Value

Epidemiological characteristics

Age in years, median (IQR)

55.7 (52.9-61.7)

57.0 (52.7-61.1)

0.662

Female sex; n (%)

35 (87.5)

34 (85.0)

0.745

Smoking

0.181

   Never smoked, n (%)

15 (37.5)

21 (52.5)

   Exsmoker, n (%)

19 (47.5)

11 (27.5)

   Active smoker, n (%)

6 (15.0)

8 (20.0)

Comorbidities

   Arterial hypertension, n (%)

10 (25.0)

9 (22.5)

0.792

   Diabetes mellitus, n (%)

2 (5.0)

4 (10.0)

0.395

   Cardiovascular disease, n (%)

3 (7.5)

2 (5.0)

0.644

   Family history of coronary artery disease, n (%)

13 (32.5)

7 (17.5)

0.121

Anthropometric characteristics

BMI (kg/m2), median (IQR)

26.7 (24.5-31.0)

27.2 (24.4-30.8)

0.758

   Obesity, n (%)

10 (26.3)

10 (27.0)

0.944

Waist circumference, (cm), median (IQR)

91.5 (83.0-108.5)

91 (84.0-102.0)

0.361

Hip circumference (cm), median (IQR)

106.5 (103.5-112.3)

105.0 (100.0-114.0)

0.308

Waist-hip ratio, median (IQR)

0.86 (0.8-0.9)

0.86 (0.8-0.9)

0.828

MET-minute, median (IQR)

495.0 (70.0-990.0)

893.0 (280.5-1188.0)

0.008

Total MEDAS score, median (IQR)

10.0 (8.0-11.0)

9.0 (8.0-11.0)

0.184

Framingham %, median (IQR)

2.6 (0.9-4.1)

1.8 (0.7-4.6)

0.501

    High risk, n (%)

0 (0.0)

0 (0.0)

1.000

    Intermediate risk, n (%)

6 (16.2)

3 (8.1)

0.286

    Low risk, n (%)

31 (83.8)

34 (91.9)

0.286

Clinical-laboratory characteristics

Progression of RA, months, median (IQR)

119 (81.2-167.9)

-

-

Diagnostic delay, months, median (IQR)

8.1 (5.6-16.7)

-

-

Erosions, n (%)

16 (40.0)

-

-

RF >10, n (%)

26 (65.0)

0 (0.0)

<0.001

ACPA >20, n (%)

31 (77.5)

0 (0.0)

<0.001

High-sensitivity CRP (mg/dl), median (IQR)

4.2 (2.7-7.4)

1.7 (0.8-3.1)

0.002

ESR (mm/h), median (IQR)

15 (9.0-26.5)

11 (6.6-18.5)

0.016

DAS28 at protocol, median (IQR)

3.06 (2.5-4.2)

-

-

   Remission-low activity, n (%)

21 (53.8)

-

-

   Moderate-high activity, n (%)

18 (46.1)

-

-

HAQ, median (IQR)

0.9 (0.2-1.6)

-

-

Synthetic DMARDs, n (%)

31 (77.5)

-

-

   Methotrexate, n (%)

23 (62.2)

-

-

   Leflunomide, n (%)

3 (8.1)

-

-

   Sulfasalazine, n (%)

3 (8.1)

-

-

   Hydroxychloroquine, n (%)

2 (5.4)

Biologic DMARDs, n (%)

21 (52.5)

-

-

    Anti TNF-α, n (%)

17 (45.9)

-

-

    Jak inhibitor, n (%)

1 (2.7)

-

-

    Anti-IL-6, n (%)

3 (8.1)

-

-

Glucocorticoid at protocol, n (%)

13 (32.5)

-

-

Glucocorticoid dose at protocol, median (IQR)

5 (5.0-5.0)

-

-

Other treatments

   Antihypertensive drugs

10 (25.0)

9 (22.5)

0.792

   ACEIs, n (%)

7 (17.5)

7 (17.5)

0.778

   ARAIIs, n (%)

3 (7.5)

2 (5.0)

0.462

   Diuretics, n (%)

5 (12.5)

8 (20.0)

0.370

   Metformin, n (%)

2 (5.0)

3 (7.5)

0.320

   Insulin, n (%)

0 (0.0)

1 (2.5)

0.320

   Other oral antidiabetic agents, n (%)

0 (0.0)

1 (2.5)

0.320

Abbreviations: RA, rheumatoid arthritis; ACPA, anti-citrullinated peptide antibodies; RF, rheumatoid factor; SD, standard deviation; MEDAS, Mediterranean Diet Adherence Survey; DAS28, 28-joint Disease Activity Score; HAQ, Health Assessment Questionnaire; CRP, C-reactive protein; ESR, erythrocyte sedimentation rate; DMARD, disease-modifying antirheumatic drug; IL-6, interleukin 6; Anti TNF, anti–tumor necrosis factor ACEI, angiotensin-converting enzyme inhibitor; ARAII, angiotensin II receptor antagonists.

Table 2. Lipid profile and carotid ultrasound in 40 patients with RA and 40 controls.

Variable

RA

n=40

Controls

n=40

RA vs Controls

p

Fasting

Postprandial

Fasting

Postprandial

Fasting

Postprandial

Fasting lipid profile

   Total cholesterol (mg/dl), median (IQR)

212.1 (187.0-234.2)

202.0 (178.0-226.2)

200.2 (176.0-227.2)

201.0 (168.1-220.5)

0.148

0.222

   LDL cholesterol (mg/dl), median (IQR)

127.0 (107.1-140.0)

110.3 (98.5-130.0)

116.5 (95.7-140.5)

108.0 (83.1-128.6)

0.229

0.203

   HDL cholesterol (mg/dl), median (IQR)

62.5 (54.7-76.5)

62.2 (52.7-72.2)

59.5 (47.7-71)

57.1 (46.2-67.2)

0.162

0.063

   Triglycerides (mg/dl), median (IQR)

82.5 (66.7-113.5)

130.0 (91.7-185.0) *

88.5 (64.5-125.7)

132.5 (108.2-210.4) *

0.823

0.913

   Chylomicrons (triglycerides), median (IQR)

14.7 (10.3-27.4)

42.3 (22.1-81.3) *

16.4 (7.5-39.8)

43.7 (31.9-84.7) *

0.644

0.225

   Chylomicrons (cholesterol), median (IQR)

9.2 (6.8-13.5)

9.2 (6.8-13.5) *

12.3 (5.3-21.9)

12.3 (5.3-21.9) *

0.544

0.613

   VLDL (triglycerides), median (IQR)

16.0 (10.1-27.7)

29.6 (15.5-41.1) *

21.0 (9.8-31.5)

24.6 (17.6-38.7) *

0.758

0.859

   VLDL (cholesterol), median (IQR)

3.6 (2.3-6.1)

3.6 (2.3-6.1) *

5.8 (2.7-9.8)

5.8 (2.7-9.8) *

0.087

0.083

   ApoB48, median (IQR)

7.4 (6.2-10.5)

14.4 (10.8-23.2) *

7.7 (5.5- 10.3)

12.1 (10.9-16.2) *

0.874

0.042

   ApoB total, median (IQR)

96.1 (84.9-104.9)

92.4 (80-103.4)

98.0 (84.3-108.3)

93.2 (77.3-102.9)

0.950

0.517

   TG/HDL ratio, median (IQR)

1.2 (0.8-2.0)

2.1 (1.3-3.5)

1.4 (0.8-2.9)

2.5 (1.6-4.1)

0.597

0.574

   ApoB48/TG ratio, median (IQR)

0.09 (0.07-0.1)

0.1 (0.7-0.1)

0.08 (0.1-0.13)

0.09 (0.7-0.1)

0.985

0.326

Fasting carbohydrate profile

   Baseline blood sugar (mg/dl), median (IQR)

78.0 (74.7-83)

80.0 (72.7-88.2)

0.843

0.277

   Homocysteine, median (IQR)

14.4 (12.8-18)

13.5 (11.4- 16.8)

0.494

Carotid ultrasound

Pathologic cIMT >p90, n (%)

10 (25.0)

9 (22.5)

0.555

Right cIMT (mm), median (IQR)

0.7 (0.6-0.8)

0.7 (0.7-1.0)

0.652

Left cIMT (mm), median (IQR)

0.66 (0.6-0.7)

0.7 (0.64-0.78)

0.353

Patients with atheromatous plaques, n (%)

7 (18.4)

8 (20.0)

0.481

* p <0.005 fasting vs postprandial value. Abbreviations cIMT, carotid intima media thickness; LDL, low-density lipoprotein; HDL, high-density lipoprotein; TG, triglycerides; VLDL, very-low-density lipoproteins.

Table 3. Baseline characteristics of patients with RA according to cIMT.

Variable

RA with IMT >p90 n=10

RA with IMT ≤p90

n=30

p Value

Age, years, median (IQR)

55.3 (48.6-68.3)

55.7 (53.3-61.6)

0.900

Female sex; n (%)

6 (60.0)

29 (96.7)

0.002

Smoking

0.818

   Never, n (%)

4 (40.0)

11 (36.7)

   Exsmoker, n (%)

4 (40.0)

15 (50.0)

   Active smoker, n (%)

2 (20.0)

4 (13.3)

Comorbidities

   Arterial hypertension, n (%)

3 (30.0)

7 (23.3)

0.673

   Diabetes mellitus, n (%)

1 (10.0)

1 (3.3)

0.442

   Cardiovascular disease, n (%)

0 (0.0)

3 (10.0)

0.298

Anthropometric characteristics

BMI (kg/m2), median (IQR)

27.1 (24.4-32.2)

26.6 (24.6-29.5)

0.700

   Obesity, n (%)

Waist circumference, (cm), median (IQR)

106.5 (87-110.7)

89 (83-103)

0.212

Hip circumference (cm), median (IQR)

106 (103.2-109.7)

106.5 (102.2-112.2)

0.941

Waist-hip index, median (IQR)

0.92 (0.83-1)

0.85 (0.81-0.91)

0.048

MET-minute, median (IQR)

247.5 (70.0-618.7)

594.0 (84.0-1064.0)

0.164

Total MEDAS, median (IQR)

10. (8.7-11.0)

10.0 (8.0-11.0)

0.824

Framingham %, median (IQR)

4.6 (1.5-13.8)

1.2 (0.6-3.8)

0.039

Clinical-laboratory characteristics

Time since diagnosis of RA, months, median (IQR)

140 (93-214.4)

113 (80-166.2)

0.138

Diagnostic delay, months, median (IQR)

9.9 (5.5-18.5)

6.9 (5.3-12.0)

0.414

Erosions, n (%)

4 (40.0)

12 (40.0)

0.473

RF >10, n (%)

7 (70.0)

21 (70.0)

1.000

ACPA >20, n (%)

8 (80.0)

22 (73.3)

0.473

High ACPA (>340), n (%)

6 (60.0)

10 (33.3)

0.036

High-sensitivity CRP (mg/dl), median (IQR)

4.4 (3.2-8.7)

3.8 (2.5-7.5)

0.221

ESR (mm/h), median (IQR)

12.0 (7.7-37.2)

15.0 (9.0-26.0)

0.839

DAS28 at protocol, median (IQR)

3.3 (2.5-3.9)

2.9 (2.5-4.2)

0.644

HAQ, median (IQR)

1.3 (0.7-1.7)

0.8 (0.2-1.6)

0.544

Synthetic DMARDs, n (%)

9 (90.0)

22 (73.0)

0.174

Methotrexate, n (%)

5 (50.0)

18 (60.0)

0.580

Biologic DMARDs, n (%)

4 (40.0)

18 (60.0)

0.271

Corticosteroids, n (%)

6 (60.0)

7 (23.3)

0.032

Abbreviations: cIMT, carotid intima media thickness.

Table 4. Lipid profile and carotid ultrasound in RA patients according to cIMT.

Variable

RA with IMT >p90 n=10

RA with IMT ≤p90

n=30

 RA with IMT >p90 vs RA with IMT ≤p90

p

Fasting

Postprandial

Fasting

Postprandial

Fasting

Postprandial

Fasting lipid profile

   Total cholesterol (mg/dl), median (IQR)

226.0 (189.5-243.2)

204.1 (184-237.2)

210.2 (187.2-225.7)

200.0 (179.2-222.2)

0.471

0.573

   LDL cholesterol (mg/dl), median (IQR)

138.2 (118.7-152.0)

110.0 (107.2-130.5)

123.5 (105.5-139.0)

110.0 (95.5-129.7)

0.266

0.647

   HDL cholesterol (mg/dl), median (IQR)

61.0 (51.2-69.7)

57.5 (49.0-65.7)

66.5 (56.5-77.5)

63.5 (53.7-73.7)

0.045

0.042

   Triglycerides (mg/dl), median (IQR)

112.0 (82.7-17.6)

195.0 (127.5-285.2)

77.5 (863.5-107.2)

116.0 (83.5-176.2)

0.014

0.033

   Chylomicrons (triglycerides), median (IQR)

33.8 (14.2-57.0) *

79.1 (25.6-167.1) *

14.1 (9.3-23.2)

32.7 (21.8-54.7)

0.066

0.045

   Chylomicrons (cholesterol), median (IQR)

12.3 (8.6-16.1)

15.6 (5.7-25.0)

8.7 (6.0-11.6)

15.3 (7.3-24.5)

0.089

0.770

   VLDL (triglycerides), median (IQR)

24.2 (21.9-39.5)

42.6 (17.3-60.0)

14.4 (9.8-26.4)

23.1 (13.1-39.7)

0.022

0.036

   VLDL (cholesterol), median (IQR)

5.3 (3.9-9.4)

8.2 (3.7-12.7)

2.8 (1.9-6.0)

3.9 (2.7-8.9)

0.028

0.046

   ApoB48, median (IQR)

8.5 (5.9-13.0) *

24.3 (15.1-27.1) *

7.7 (5.5- 10.3)

13.5 (10.5-18.2)

0.186

0.017

   ApoB total, median (IQR)

102.7 (94.4-115.1)

98.0 (87-110.8)

94 (82.6-104.3)

91.7 (76.1-102.6)

0.141

0.100

Increased postprandial blood lipids

   Triglycerides (mg/dl), median (IQR)

73.2 (24.0-134.5)

39.9 (17.2-68.0)

0.122

   Chylomicrons (triglycerides), median (IQR)

47.4 (14.6-124.1)

21.5 (10.2-37.7)

0.045

   VLDL (triglycerides), median (IQR)

12.9 (6.4-20.7)

8.0 (3.0-16.2)

0.424

   ApoB48, median (IQR)

12.3 (10.8-14.3)

6.7 (3.4-8.6)

0.002

* p <0.005 fasting vs postprandial value. Abbreviations: cIMT, carotid intima media thickness; LDL, low-density lipoprotein; HDL, high-density lipoprotein; TG, triglycerides; VLDL, very-low-density lipoprotein

  1. Why secondary Sjögren syndrome was not an exclusion criteria?

R: We did not exclude secondary Sjögren syndrome because it is considered by most authors as part of the same spectrum of rheumatoid arthritis in most patients.

Rferences:

Anaya JM, Rojas-Villarraga A, Mantilla RD, Arcos-Burgos M, Sarmiento-Monroy JC. Polyautoimmunity in Sjögren syndrome. Rheum Dis Clin North Am 2016;42:457–72.

Shiboski SC, Shiboski CH, Criswell L et al. American College of Rheumatology classification criteria for Sjogren’s syndrome: A data-driven, expert consensus approach in the Sjögren’s International Collaborative Clinical Alliance cohort. Arthritis Care Res (Hoboken) 2012; 64; 475–87.

Reviewer 3 Report

Interesting study, requires much effort yet needs minor grammar/spelling check and some comments:

a. suggested edits:

line 45: multiple morbidities

line 47: to both

line 71: plaque burden/ presence

line 77: remove 1st (and), use coma instead

b. comments:

line 91, if possible to be more specific and give clarification of disease symptoms

line 95, sample taken from all patients or all study population ( patients and controls), please  double check when to use patients, controls or study population.

line 121, how many fasting hours were required before checking blood lipid levels? 

line 194, please make sure that median and P value match those in the table, unless it's a different method of calculation.  since median subtraction of apob48 from table 2 doesn't give those results.

I have some conservation regarding the selection and randomaization of controls. Also about external validity. 

Author Response

Revisor 3

Comments and Suggestions for Authors

Interesting study, requires much effort yet needs minor grammar/spelling check and some comments:

  1. suggested edits:

R: We appreciate your comments and following the recommendation offered by the Reviewer we have changed this point.

line 45: multiple morbidities

R: The authors changed in line 45 “It is associated with premature death and multiple morbidities”

line 47: to both

R: The authors changed in line 47 “Accelerated atherosclerosis in patients with RA is due to both the presence of traditional cardiovascular risk factors and to non-traditional cardiovascular risk factors, including systemic inflammation and dyslipidemia”

line 71: plaque burden/ presence

R: The authors changed in line 71: “The objective of the present study was to analyze the association between postprandial lipid values (i.e., ApoB48 levels after a mixed breakfast) and subclinical atherosclerosis measured as cIMT and plaque presence in patients with RA.”

line 77: remove 1st (and), use coma instead

R: The authors changed in line 77: “The study was carried out at IBIMA, in the Rheumatology Department of Hospital Regional Universitario de Málaga (HRUM), in the Lipid and Atherosclerosis Laboratory of Universidad de Málaga (UMA) and the Center for Medical and Health Research (CIMES), Spain”

  1. comments:

line 91, if possible, to be more specific and give clarification of disease symptoms

R: We added in line 91 “The controls were volunteers aged more than 16 years with no inflammatory or autoimmune diseases or symptoms that would lead us to suspect these diseases (i.e. joint inflammation, inflammatory pain, morning stiffness, etc.). The exclusion criteria were the same as for the patients. The controls were selected at random from a health center in the catchment area of the hospital. Controls and patients were matched by age and sex."

line 95, sample taken from all patients or all study population (patients and controls), please  double check when to use patients, controls or study population.

R: The authors changed in line 95: “The study population then had a mixed breakfast in the hospital cafeteria (milk, cured ham, cheese, olive oil, and bread).”

line 121, how many fasting hours were required before checking blood lipid levels? 

R: The study population fasted 8 hours

We added in line 125-127: “ApoB48 and total ApoB levels were measured in plasma after 8 hours fasting and 4 hours after the meal using a commercially available ELISA approach (Shibayagi Co Ltd, Ishihara, Japan)”

line 194, please make sure that median and P value match those in the table, unless it's a different method of calculation.  since median subtraction of apob48 from table 2 doesn't give those results.

R: The authors apologize for this mistake in text. The data in the table is correct.

We have changed the text in line 207-208: “The increase in ApoB48 was significantly greater in patients than in controls (median [IQR], 14.4 [10.8-23.2] vs 12.1 [10.9-16.2]; p=0.042).”

I have some conservation regarding the selection and randomization of controls. Also, about external validity. 

R: Our patients were consecutively included from our RA consultations whilst controls were chosen by simple randomization of a listing and then they were invited to participate. Our sample of patients is the usual one in routine clinical practice, patients undergoing the same treatments and comorbidities as in clinical practice, so we think that these results can be generalized to other patients with RA.

Round 2

Reviewer 2 Report

No further comments